

# Coral-algal interactions at Weizhou Island in the northern South China Sea: variations by taxa and the exacerbating impact of sediments trapped in turf algae

Zhiheng Liao[1,2,3], Kefu Yu[1,2,3], Yinghui Wang[1,2,3], Xueyong Huang[1,2,3,4] and Lijia Xu[1,2,3]

[1] Coral Reef Research Center of China, Guangxi University, Nanning, Guangxi, China
[2] Guangxi Laboratory on the Study of Coral Reefs in the South China Sea, Guangxi University, Nanning, Guangxi, China
[3] School of Marine Sciences, Guangxi University, Nanning, Guangxi, China
[4] School of Forestry, Guangxi University, Nanning, Guangxi, China

Corresponding author
Kefu Yu, kefuyu@scsio.ac.cn

## ABSTRACT

Competitive interactions between corals and benthic algae are increasingly frequent on degrading coral reefs, but the processes and mechanisms surrounding the interactions, as well as the exacerbating effects of sediments trapped in turf algae, are poorly described. We surveyed the frequency, proportion, and outcomes of interactions between benthic algae (turf algae and macroalgae) and 631 corals (genera: *Porites*, *Favites*, *Favia*, *Platygyra*, and *Pavona*) on a degenerating reef in the northern South China Sea, with a specific focus on the negative effects of algal contact on corals. Our data indicated that turf algae were the main algal competitors for each surveyed coral genus and the proportion of algal contact along the coral edges varied significantly among the coral genera and the algal types. The proportions of algal wins between corals and turf algae or macroalgae differed significantly among coral genera. Compared to macroalgae, turf algae consistently yielded more algal wins and fewer coral wins on all coral genera. Amongst the coral genera, *Porites* was the most easily damaged by algal competition. The proportions of turf algal wins on the coral genera increased 1.1–1.9 times in the presence of sediments. Furthermore, the proportions of algal wins on massive and encrusting corals significantly increased with the combination of sediments and turf algae as the algal type. However, the variation in proportions of algal wins between massive and encrusting corals disappeared as sediments became trapped in turf algae. Sediments bound within turf algae further induced damage to corals and reduced the competitive advantage of the different coral growth forms in their competitive interactions with adjacent turf algae.

## INTRODUCTION

Coral reefs around the world, including those of the South China Sea, have been undergoing rapid degradation due to anthropogenic stressors and changes in their

natural environment (*Yu, 2012*). Macroalgae and turf algae generally tend to increase in abundance in degraded reefs (*Nugues & Bak, 2006*; *Haas, El-Zibdah & Wild, 2010*) and develop intense competitive interactions with the remaining corals (*Hughes, 1994*; *McCook, Jompa & Diaz-Pulido, 2001*; *Bellwood et al., 2004*). This is particularly pronounced in reefs that experience frequent human activity (*Barott et al., 2012*; *Brown et al., 2017*). Indeed, phase shifts from coral to algal domination in reefs have been observed with increasing frequency as a result of overfishing (*Hughes, 1994*), eutrophication (*Littler, Littler & Brooks, 2006*), sediment deposition (*Goatley & Bellwood, 2013*; *Goatley et al., 2016*), and global climate change (*Hoegh-Guldberg et al., 2007*).

The interaction between benthic algae and scleractinian corals can be a major determinant of reef community structure and composition, particularly in degrading reefs that are dominated by macroalgae (*McCook, Jompa & Diaz-Pulido, 2001*). The outcomes of such interactions mainly depend on the species involved (*McCook, Jompa & Diaz-Pulido, 2001*; *Nugues & Bak, 2006*), as the competitive mechanisms and strengths of algae and the resistance abilities of corals varies among species (*Jompa & McCook, 2003*; *Nugues & Bak, 2006*; *Bonaldo & Hay, 2014*). Moreover, polyp size (hypothesized by *McCook, Jompa & Diaz-Pulido (2001)*), coral colony size (*Barott et al., 2012*; *Ferrari, Gonzalez-Rivero & Mumby, 2012*; *Brown et al., 2017*), coral growth form (*Swierts & Vermeij, 2016*), and environmental factors (*Vermeij et al., 2010*) are also important determinants of the outcomes of coral-algal interactions. *Barott et al. (2012)* found that large and small coral colonies were more successful in competing against algae than mid-sized corals because they were able to allocate more energy to the competition. Different coral growth forms can also influence the nature of competitive interactions between coral and algae (*Swierts & Vermeij, 2016*). Different growth forms use either one of two, or both, strategies to compete against algae: the "escape in height" strategy, in which corals invest energy in vertical growth to escape algal competition, and the "direct confrontation" strategy, in which corals directly fight off algal invasion (*Meesters, Wesseling & Bak, 1996*; *McCook, Jompa & Diaz-Pulido, 2001*; *Swierts & Vermeij, 2016*). In addition to these strategies, *McCook, Jompa & Diaz-Pulido (2001)* suggested that coral species-specificity can affect the outcomes of their interactions with algae. This interactive process depends more on the properties (e.g., species, functional groups, etc.) of the algae than of the coral (*McCook, Jompa & Diaz-Pulido, 2001*; *Jompa & McCook, 2003*).

On coral reefs, macroalgae and turf algae are the main competitors against corals for space (*McCook, Jompa & Diaz-Pulido, 2001*; *Vermeij et al., 2010*). With regard to coral-algal interactions, macroalgae are the most frequently researched and have been shown to inhibit nearby coral growth (*Jompa & McCook, 2003*; *Titlyanov, Yakovleva & Titlyanova, 2007*), recruitment (*Birrell et al., 2008*; *Vermeij et al., 2009*) and fecundity (*Foster, Box & Mumby, 2008*). Crustose coralline algae (CCA) are generally thought to have positive or minimal negative effects on corals (*Negri et al., 2001*; *Harrington et al., 2004*; *Vermeij & Sandin, 2008*). Compared to macroalgae and CCA, turf algae grow quickly and can pre-emptively colonize available space (*McCook, Jompa & Diaz-Pulido, 2001*; *Littler, Littler & Brooks, 2006*). Turf algae interact most frequently

with corals on degraded reefs (*Littler, Littler & Brooks, 2006*; *Haas, El-Zibdah & Wild, 2010*) and can cause hypoxia (*Smith et al., 2006*; *Wangpraseurt et al., 2012*; *Gregg et al., 2013*; *Haas et al., 2013*; *Roach et al., 2017*), physical damage (*McCook, Jompa & Diaz-Pulido, 2001*), bleaching (*Titlyanov, Yakovleva & Titlyanova, 2007*; *Rasher & Hay, 2010*; *Rasher et al., 2011*), and disease (*Barott & Rohwer, 2012*) in adjacent corals. Turf algae have also been seen to have critical negative effects on the settlement and recruitment of coral larvae (*Birrell, McCook & Willis, 2005*; *Birrell et al., 2008*). In addition, algal turfs often accumulate a variety of sediments on the reef substrate (*Purcell, 2000*; *Birrell, McCook & Willis, 2005*), which exacerbate the stress and mortality that they cause in underlying corals and CCA (*Steneck, 1997*; *Nugues & Roberts, 2003*; *Cetz-Navarro et al., 2013*). The harmful effects of sediments are influenced by water flow, for example, low flow rates can increase the rates of sediment accumulation (*Gowan, Tootell & Carpenter, 2014*). When sediments are bound within turf algae, grazing by herbivores may also be inhibited (*Wilson & Bellwood, 1997*; *Bellwood & Fulton, 2008*; *Goatley & Bellwood, 2012*), which further enables the growth of algal turfs and hinders the settlement of coral larvae (*Birrell, McCook & Willis, 2005*; *Goatley et al., 2016*).

Although various researchers have conducted sound investigations into the influences of different algae on corals, their field surveys often do not consider the diversity of traits associated with the competitors or the environmental factors that could influence the outcome of coral-algal interactions. Field surveys on the influence of coral traits (e.g., genus, growth form, polyp size, etc.) and sediments on coral-algal interactions may help to improve our understanding of the complex processes involved in these competitive interactions. Such insights may be helpful in the selection of corals for use as coral transplants in the restoration of degraded coral reefs that have become dominated by algae.

In the present study, we conducted field surveys to investigate the influence of turf algae and macroalgae (*Lobophora variegata* and *Bryopsis pennata*) on five common coral genera (*Pavona*, *Porites*, *Favites*, *Favia*, and *Platygyra*) on the coral reef of Weizhou Island, northern South China Sea. The study objectives were: (1) to survey the frequency of coral-algal contact, as well as the composition and proportion of contiguous algae along the coral edges; (2) to determine the outcomes of the competitive interactions between the common corals and dominant algae; and (3) to determine the impact of accumulated sediments on the negative effects of turf algae (i.e., algal wins) on corals. Our central hypothesis was that sediment deposition would increase the competitiveness of turf algae in coral-algal interactions, enable them to overgrow corals, and reduce the competitive advantage of the coral growth forms.

## MATERIALS AND METHODS

### Study site

The survey was conducted in October 2015 (autumn) at Weizhou Island (21°03′N, 109°07′E); the largest and youngest volcanic island in the Chinese coastal areas, located 48 km from the mainland coastline in the Beibu Gulf in the northern South China Sea

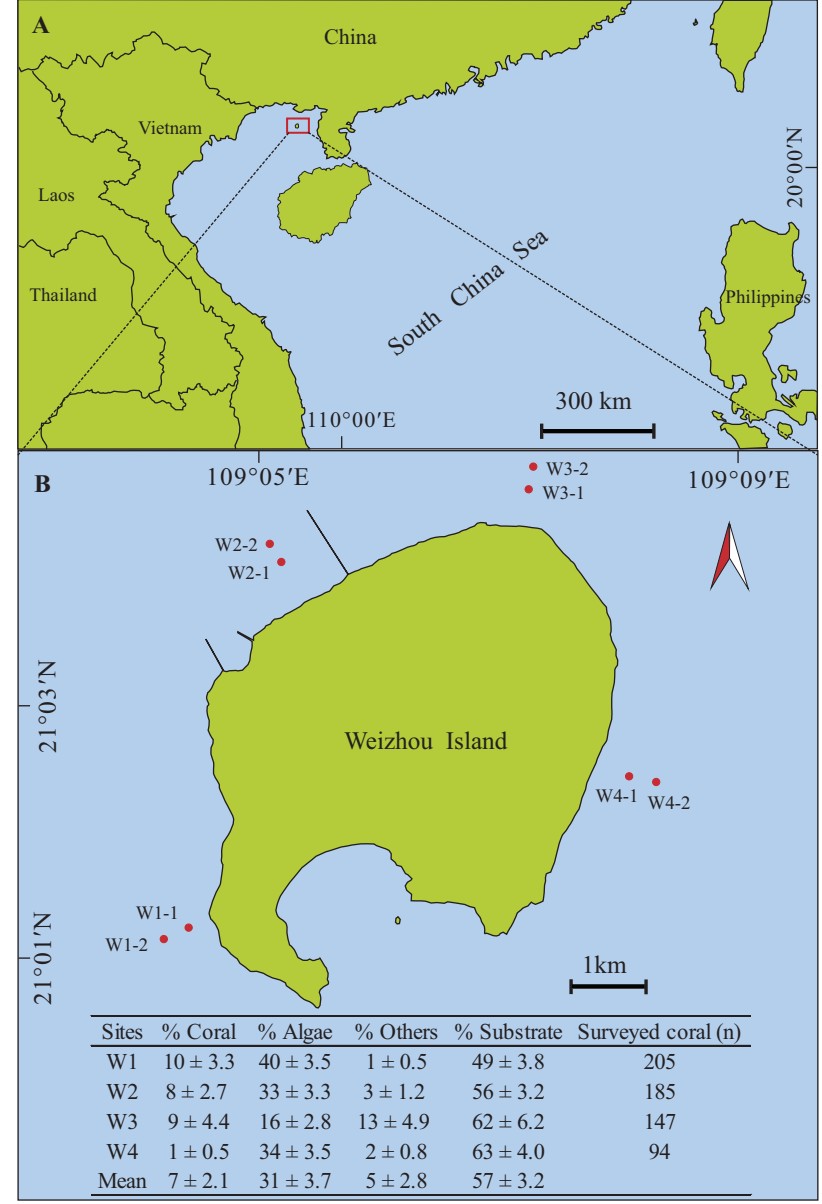

**Figure 1 Maps of Weizhou Island, Beibu Gulf, and study sites.** (A) Map of Weizhou Island in the Beibu Gulf of the South China Sea. (B) The eight transects are marked with red dots; the table inset shows the cover of biological and non-biological substrates and the number of surveyed coral colonies at each site.

(Fig. 1A). Weizhou Island has experienced a dramatic decline in its coral reefs since the 1990s: from 80% coral cover to an estimated current live coral cover of less than 10% (*Wang, Yu & Wang, 2016*). The rapid development of tourism, fishing, aquaculture, and other human activities in recent years have likely had detrimental effects on the coral reef ecosystem. In addition, long-term investigations have indicated that fish populations have been at low levels for a long time (*Chen et al., 2016*). However, no recent surveys have been conducted on the benthic algae of the Weizhou coral reef.
## Benthic cover

Field research was conducted with permission from the School of Marine Sciences, Guangxi University. Coverage of hard corals and benthic algae and coral-algal interactions were surveyed along transects at four sites on the reef flat of Weizhou Island. The distance between sites was greater than one km (Fig. 1B). Two 100 m parallel transects, separated by over 100 m, were installed at each of the four sites at a depth of three to seven m; that is, a total of eight transects were surveyed. For each transect, a 100 m fiberglass measuring tape was fixed to the reef flat and sampling quadrats (0.5 × 0.5 m) were placed every 10 m along the transect.

An OLYMPUS TG-4 camera (effective pixel density: 12 million pixels) was used to photograph each quadrat. Benthic cover and composition were analyzed using Coral Point Count with Excel extensions software that used 50 randomly-placed points within each frame (Preskitt, Vroom & Smith, 2004; Kohler & Gill, 2006). Reef benthic cover was calculated from 11 quadrats along each of the eight transects, that is, n = 88 quadrats. Hard corals were identified to the genus level, macroalgae were identified to the species or genus level, and CCA and turf algae were classified as individual functional groups. We also identified other benthic organisms, such as soft corals and sponges. Non-biological substrates were classified as sand, rock, rubble, and dead coral.

## Frequency of contact, proportion, and outcomes of coral-algal interactions

Videos were also recorded along each transect (as previously described) using an OLYMPUS TG-4 camera, held 15–25 cm above the benthos by a SCUBA diver at a swimming speed of four m min$^{-1}$. The frequency and proportion of coral-algal interactions (i.e., algal contacts along coral edges) were assessed based on screenshots of the transect videos. We derived top-view photographs (screenshots) of the common corals (*Porites*, *Favites*, *Favia*, *Platygyra*, and *Pavona* genera). We ensured that these photographs clearly reflected information regarding the coral genus, algal type, and algal contacts along the coral edge (Fig. 2), and that the coral colony photographs revealed over half of the coral edge. We also ensured that part of the tape measure was visible alongside or above the measured corals in these photographs for the accurate quantification of coral edge perimeters and identification of the outcomes of coral-algal interactions (Fig. 2E). Using these photographs, we carefully investigated and recorded features of the coral colonies, that is, coral genus, growth form, and perimeter; we visually examined the edges of the corals for the presence or absence of turf algae and macroalgae; and we quantified the proportion of coral-algal interactions. CCA were ignored as they had minimal interactions with the corals. Using the software ImageJ 1.50i (Abramoff, Magalhaes & Ram, 2004), we determined the type and proportion of coral-algal interactions, and we recorded the contact length of each species and functional group of algae at each coral edge (Fig. 2E). The percentage of contact of each algal species around the coral was calculated by dividing the contact length of each algal taxon by the total perimeter of the coral. The outcomes of coral-algal interactions were visually evaluated from the interface (Figs. 2A and 2B) and classified as coral win, algal win,

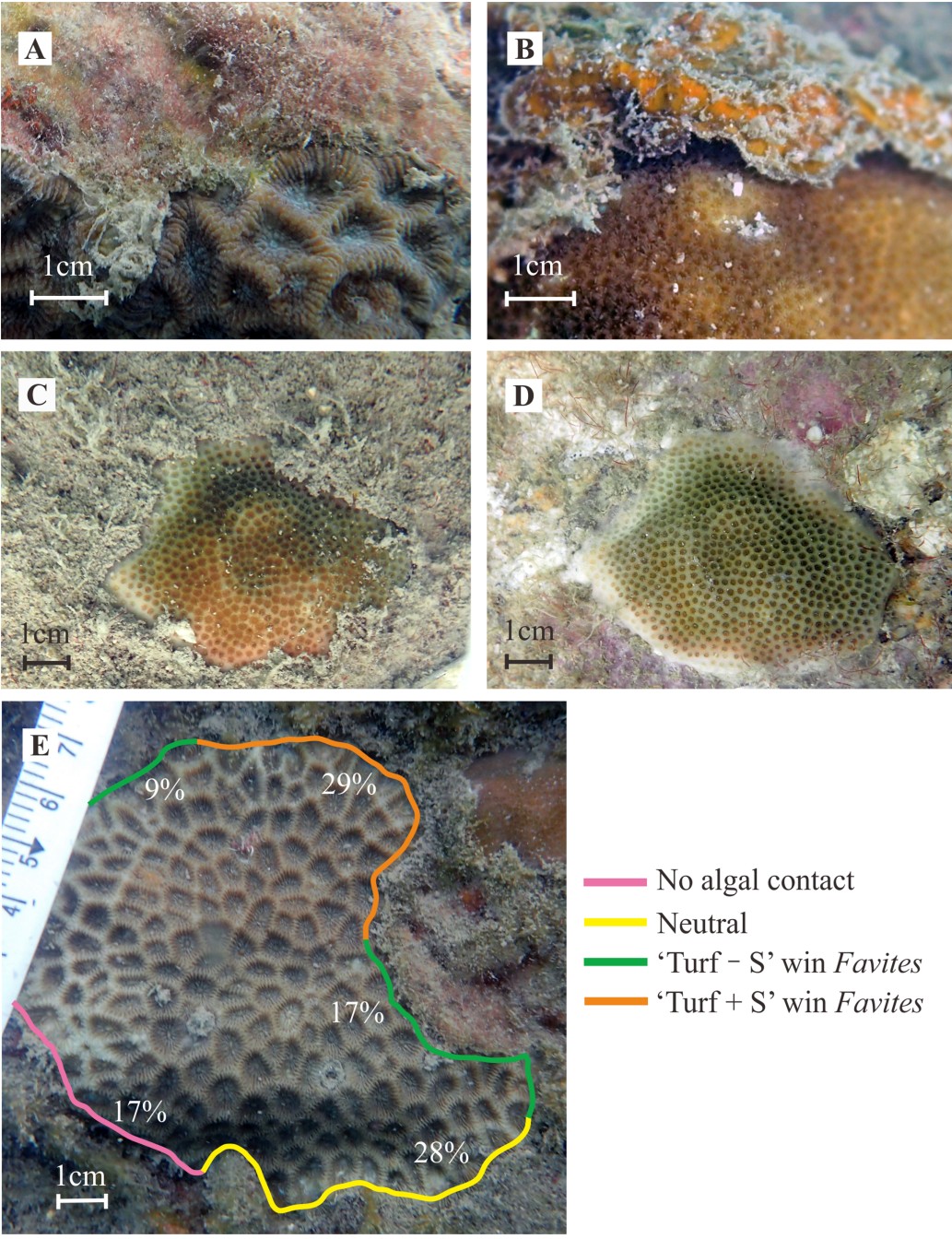

**Figure 2 Examples of coral-algal interactions and the effects of sediments bound within turf algae.** (A) and (B) Typical interaction interfaces of the *Favites* and *Porites* genera and algae (turf algae and *L. variegata*). (C) *Porites* colony surrounded by turf algae with massive sediments trapped in sparse turf algae, a widespread phenomenon at the surveyed reef; (D) the same *Porites* colony showing bleached tissue after sediments were removed using a burst of air. (E) An example of the interaction outcomes between the genus *Favites* and turf algae. "Turf + S" indicates turf algae with sediment; "Turf − S" indicates turf algae without sediment. Photo credit: Zhiheng Liao.

or neutral outcome. An outcome was scored as "coral win" if the coral had grown taller than the contiguous algae or had damaged the adjoining algal tissue. An outcome was scored as "algal win" if the coral edge showed signs of tissue necrosis, discoloration, bleaching, or if the algae had overgrown the coral colony (*Barott et al., 2012*; *Swierts & Vermeij, 2016*). An outcome was scored as "neutral" if neither the coral nor the algae caused damage or overgrowth at their interaction interface (see Fig. 2 in *Barott et al., 2012*). The proportions of algal wins were calculated for each coral by dividing the length of damage from the algae by the total length of contact of each type of algae along the coral edge. The proportions of coral wins and neutral outcomes were calculated using the same equation with respective substitutes. Outcomes were grouped by coral genera and colony growth forms, that is, coral colonies were classified as one of three growth forms: encrusting (colonies encrusting the substrate without upward growth), massive (solid colonies with similar size in all directions), and upright (colonies growing upward with a small base fixed to the substrate) based on the growth morphology of the dominant surveyed coral genera. Some coral genera, including *Porites*, *Favites*, *Favia*, and *Platygyra*, were found in both encrusting and massive growth forms, while the *Pavona* genus was found only in the upright growth form. The polyp sizes of each coral genera were classified as being smaller than two mm (i.e., *Porites* and *Pavona* genera) or larger than five mm (i.e., *Favites*, *Favia*, and *Platygyra* genera).

## Determining the effects of accumulated sediments

Even a small amount of turf algae can collect a large quantity of sediment on near-shore reef substrata (Figs. 2C and 2D). Sediments were identified using magnified (one to two times) photographs. Gray sediments sharply contrasted the colors (red, green, and brown) of the turf algae, and their presence or absence in the turf algae was recorded non-quantitatively. Turf algae that interacted with corals were divided into two categories, namely "turf algae with sediment" (Turf + S) and "turf algae without sediment" (Turf − S). We compared the proportions of algal wins that resulted from encounters between the coral colonies and "Turf + S", and between coral colonies and "Turf − S".

## Statistical analyses

Homogeneity of variances was tested on all data using Levene's test. Parametric and non-parametric analyses were applied to data that were homoscedastic and non-homoscedastic, respectively. The proportions of algal contacts (i.e., coral-algal interactions) were analyzed for differences between coral genera and algal types (e.g., macroalgae, "Turf + S", and "Turf − S") using a two-way ANOVA, with the coral genus and algal type as fixed factors. Differences in the proportions of a specific competitive outcome (e.g., coral win, algae win, or neutral) among coral genera were quantified using a Kruskal–Wallis test followed by a Student-Newman-Keuls (SNK) post hoc comparison. To assess the differences in the proportions of algal wins, a two-way ANOVA was used (with the type of turf algae or macroalgae genus and coral genus as fixed factors), followed by an SNK test. To assess the differences in the proportions of algal wins and the similarity among interaction groups (including: massive corals vs. "Turf − S", massive corals vs. "Turf + S",

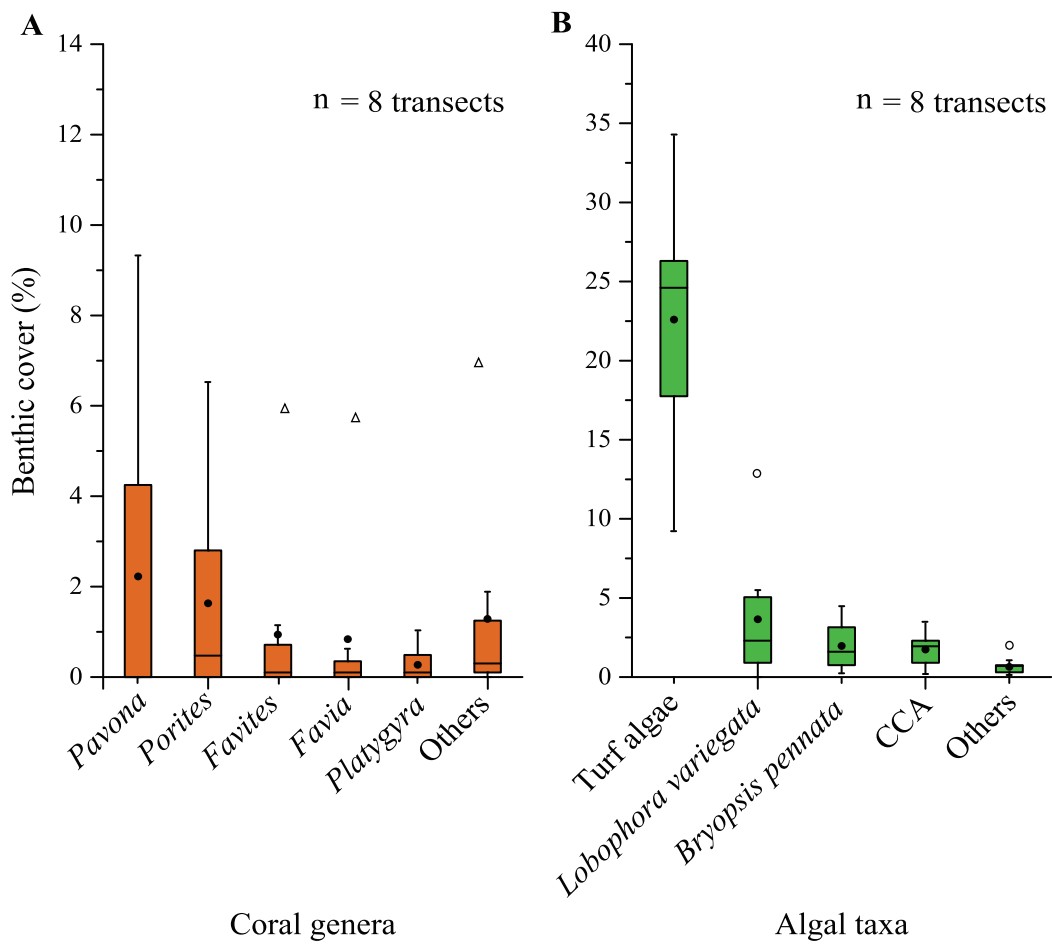

**Figure 3 Percentage of coral (A) and algal (B) cover on the coral reef of Weizhou Island.** The bar within each box represents the median; the two bars above and below each box represent the upper and lower 25% quartiles, respectively; the whiskers represent the minimum and maximum values; "•" represents the mean value; "○" indicates the data > the 1.5 interquartile range; "△" indicates the data > the 3.0 interquartile range. CCA, crustose coralline algae.

encrusting corals vs. "Turf − S", and encrusting corals vs. "Turf + S"), a two-way ANOVA (with the type of turf algae and coral growth form as fixed factors) was used, followed by an SNK test. All statistical analyses were conducted using IBM SPSS Statistics 19 software and $p < 0.05$ were considered statistically significant.

# RESULTS

## Benthic cover

The benthos of Weizhou Island consisted of 7% hard coral, 31% benthic algae, 5% other biological substrates, and the residual 57% consisted of non-biological substrates (Fig. 1B). The five common coral genera, *Pavona*, *Porites*, *Favites*, *Favia*, and *Platygyra* accounted for 31%, 23%, 13%, 11%, and 4% of the total hard coral cover, respectively (Fig. 3A). The major algal functional group was turf algae, which made up 23% of the total benthic cover. Macroalgae contributed relatively little to the
**Table 1 Frequency of coral-algal contact, and composition and proportion of algae along the coral edges.**

| Genus | Surveyed colonies (n) | Polyp size (mm) | % Colonies in contact with algae | % No algal contact | % Turf algal contact | | % Macroalgal contact |
|---|---|---|---|---|---|---|---|
| | | | | | Turf − S | Turf + S | |
| *Porites* | 211 | <2 | 86 ± 2.4 | 47 ± 1.9 | 24 ± 1.7 | 23 ± 1.9 | 6 ± 0.8 |
| *Favites* | 192 | >5 | 80 ± 5.5 | 53 ± 1.5 | 17 ± 1.5 | 25 ± 1.8 | 5 ± 0.7 |
| *Favia* | 70 | >5 | 85 ± 2.7 | 59 ± 2.8 | 18 ± 2.5 | 20 ± 3.1 | 3 ± 0.9 |
| *Platygyra* | 45 | >5 | 64 ± 7.8 | 67 ± 4.1 | 19 ± 3.3 | 10 ± 3.2 | 4 ± 1.2 |
| *Pavona* | 113 | <2 | 61 ± 3.1 | 73 ± 2.2 | 15 ± 1.5 | 3 ± 0.9 | 10 ± 1.8 |
| Mean value | | | | 56 ± 1.0 | 19 ± 0.9 | 19 ± 1.0 | 6 ± 0.5 |

Note:
Turf algae were divided into "Turf + S" (turf algae with sediments) and "Turf − S" (turf algae without sediments).

**Table 2 Results of the two-way ANOVA test for the effects of coral genera and algal types on the proportions of algal contacts along coral edges.**

| Source of variation | df | MS | F | p |
|---|---|---|---|---|
| Proportion of algal contact | | | | |
| Coral genus | 4 | 0.47 | 11.99 | <0.0001 |
| Algal type | 2 | 2.18 | 55.22 | <0.0001 |
| Coral genus × Algal type | 8 | 0.45 | 11.36 | <0.0001 |

benthic coverage, at 4% *L. variegata* and 2% *B. pennata*. Other algae contributed ~2% to the total benthic coverage (Fig. 3B).

## Algal composition, contact frequency, and proportion around the coral edges

In total, 631 coral colonies with a combined perimeter of 25,716 cm were measured in the surveyed transects. Amongst all genera, *Porites* was the most frequently in contact with algae (86% ± 2.4% of colonies; mean ± SE), followed by *Favia* (85% ± 2.7%) and *Favites* (80% ± 5.5%; Table 1). The proportions of algal contacts varied significantly with coral genera and with algal types, and a significant interaction was found between coral genus and algal type (two-way ANOVA, coral genera: $F = 11.99$, $df = 4, 1,889$, $p < 0.0001$; algal types: $F = 55.22$, $df = 2, 1,889$, $p < 0.0001$; interaction: $F = 11.36$, $df = 8, 1,889$, $p < 0.0001$; Table 2). The mean percentage of coral edges in contact with algae was higher for turf algae (38% ± 1.0%, which consisted of 19% ± 0.9% "Turf − S" and 19% ± 1.0% "Turf + S") than for macroalgae (6% ± 0.5%, $n = 631$; Table 1). The *Porites* genus was the most frequently engaged in algal contact ($n = 211$, 53% ± 1.9%), while the *Pavona* genus was the least frequently engaged in algal contact ($n = 113$, 27% ± 2.2%; Table 1).

## Interactions between corals and algae

The proportions of algal wins between the corals and turf algae were significantly higher in interactions with the *Porites* genus (which have a small polyp size), that is, 76% ± 1.7% (mean ± SE) of the interaction outcomes (65% algal overgrowth and 11% coral

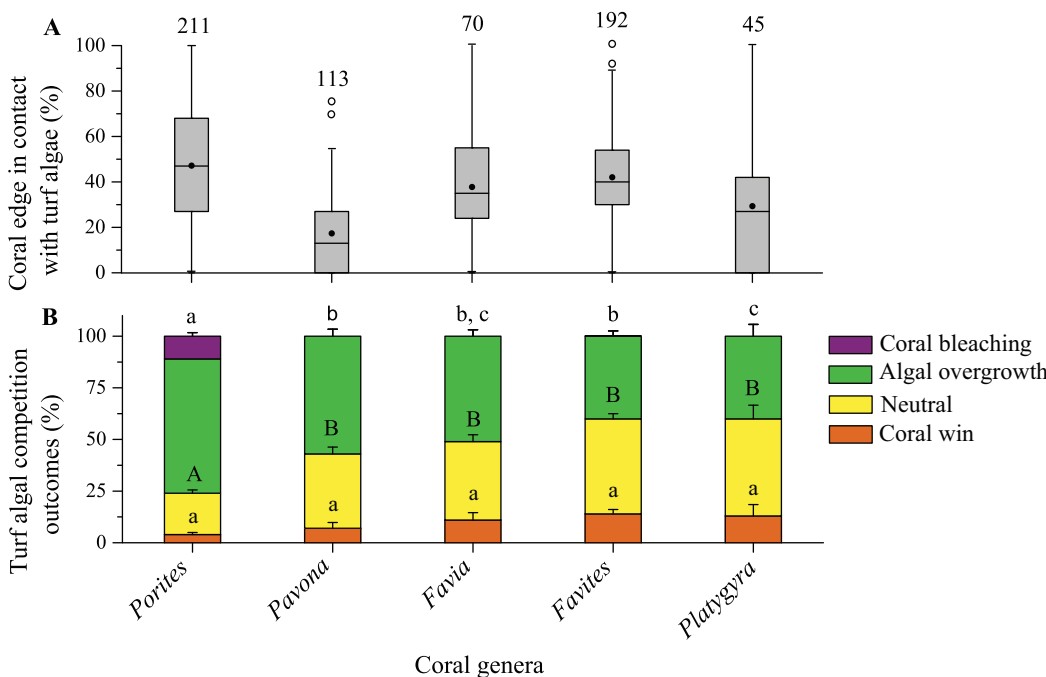

**Figure 4 Competitive outcomes between corals and turf algae.** (A) Proportions of coral edges in contact with turf algae. (B) Competitive outcomes between coral and turf algae, where green (algal overgrowth) and purple (coral bleaching) indicate the proportions of algal wins, yellow indicates the proportions of neutral outcomes, and orange indicates the proportions of coral wins. The numbers indicate the number of coral colonies included in the analyses. Similar letters (uppercase or lowercase) above each set of bars indicate significant differences (assessed by a Kruskal–Wallis test) in post hoc comparisons for a specific competitive outcome among coral genera (SNK test, $p < 0.05$).

bleaching), than with any other coral genera with large polyp sizes (Kruskal–Wallis, $X^2 = 147.65$, d$f = 4$, $p < 0.0001$; Fig. 4B). Proportions of turf algal wins in interactions with *Platygyra*, *Favites*, and *Favia* genera (with large polyp sizes) ranged from 40% ± 5.7% to 51% ± 3.1% but did not significantly differ among these genera (Fig. 4B). In these genera, algal overgrowth was the major contributor to algal wins, with few instances of coral bleaching occurring. Proportions of neutral outcomes differed significantly among coral genera, with this outcome being significantly less occurred in the *Porites* genus than in any other coral genera (Kruskal–Wallis, $X^2 = 68.27$, d$f = 4$, $p < 0.0001$; Fig. 4B). There were significant differences in the proportions of coral wins in coral-turf algal interactions among the coral genera (Kruskal–Wallis, $X^2 = 15.43$, d$f = 4$, $p = 0.0039$; Fig. 4B). Corals with small polyp sizes (<2 mm), such as the *Porites* and *Pavona* genera, were least successful in turf algal interactions and accounted for 4% ± 1.0% and 7% ± 2.8% of the coral wins, respectively (Fig. 4B).

Proportions of macroalgal wins varied significantly among coral genera (Kruskal–Wallis, $X^2 = 22.93$, d$f = 4$, $p = 0.0001$; Fig. 5B). Of the five coral genera, *Porites* was the most susceptible to damage by macroalgae, with macroalgal wins accounting for 64% ± 5.5% (mean ± SE) of their interaction outcomes (Fig. 5B). The proportions of macroalgal wins in the *Favia*, *Pavona*, *Platygyra*, and *Favites* genera ranged from 22% ± 4.0% to 51% ± 10.4%

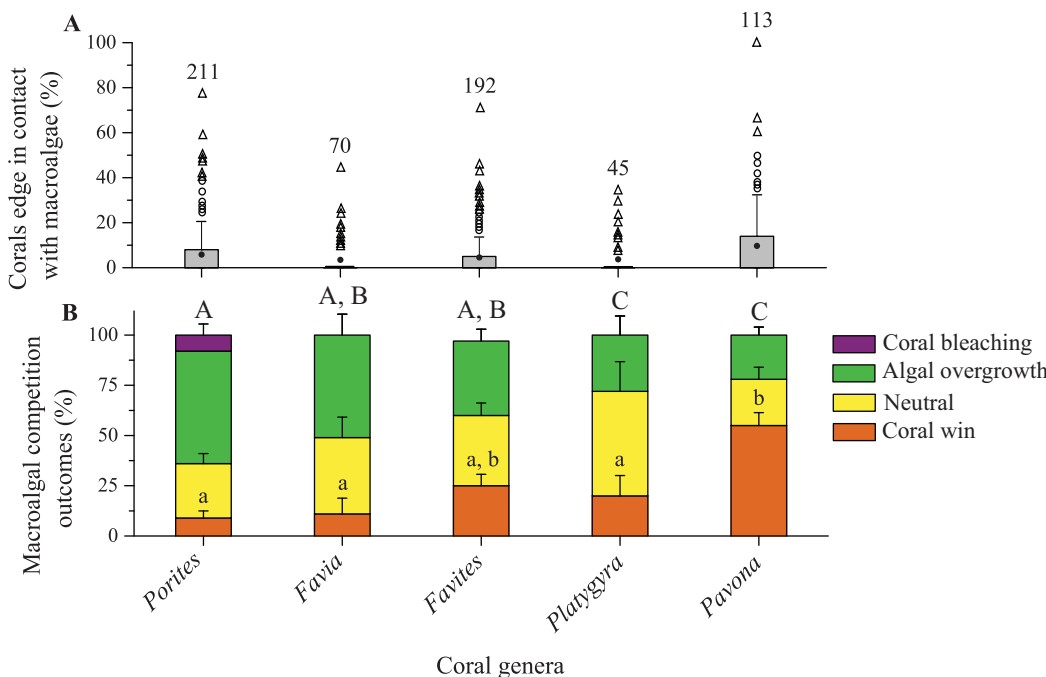

**Figure 5 Competitive outcomes between corals and macroalgae.** (A) Proportions of coral edges in contact with macroalgae. (B) Competitive outcomes between corals and macroalgae, where green (algal overgrowth) and purple (coral bleaching) indicate the proportions of algal wins, yellow indicates the proportions of neutral outcomes, and orange indicates the proportions of coral wins. The numbers indicate the number of coral colonies included in the analyses. Similar letters (uppercase or lowercase) above each set of bars indicate significant differences (assessed by a Kruskal–Wallis test) in post hoc comparisons for a specific competitive outcome among coral genera (SNK test, $p < 0.05$).

(Fig. 5B). The proportions of neutral outcomes in coral-macroalgal interactions did not significantly differ among coral genera (Kruskal–Wallis, $X^2 = 6.80$, d$f = 4$, $p = 0.147$; Fig. 5B). Corals were more successful in outcompeting macroalgae, with proportions of coral wins ranging from 9% ± 3.5% to 55% ± 6.4% (Kruskal–Wallis, $X^2 = 37.65$, d$f = 4$, $p < 0.0001$; Fig. 5B).

## Negative effects of macroalgae and turf algae in contact with corals

Proportions of turf algal wins varied among the types of turf algae and the coral genera, but there were no interactions between the type of turf algae and coral genus (two-way ANOVA, types of turf algae: $F = 30.15$, d$f = 1$, $655$, $p < 0.0001$; coral genera: $F = 57.70$, d$f = 4$, $655$, $p < 0.0001$; interaction: $F = 1.57$, d$f = 4$, $655$, $p = 0.1810$; Fig. 6A). The *Porites* genus showed significantly higher proportions of turf algal wins than other coral genera, regardless of whether the turf algae contained sediments (Fig. 6A). In all coral genera, proportions of algal wins in "Turf + S" competitions were 1.1–1.9 times higher than in "Turf − S" competitions (Fig. 6A).

In contrast, macroalgae had fewer proportions of algal wins than turf algae. Similar to turf algae, their proportions of algal wins varied with macroalgae and coral taxa, but no interactions were found between macroalgal species and coral genus
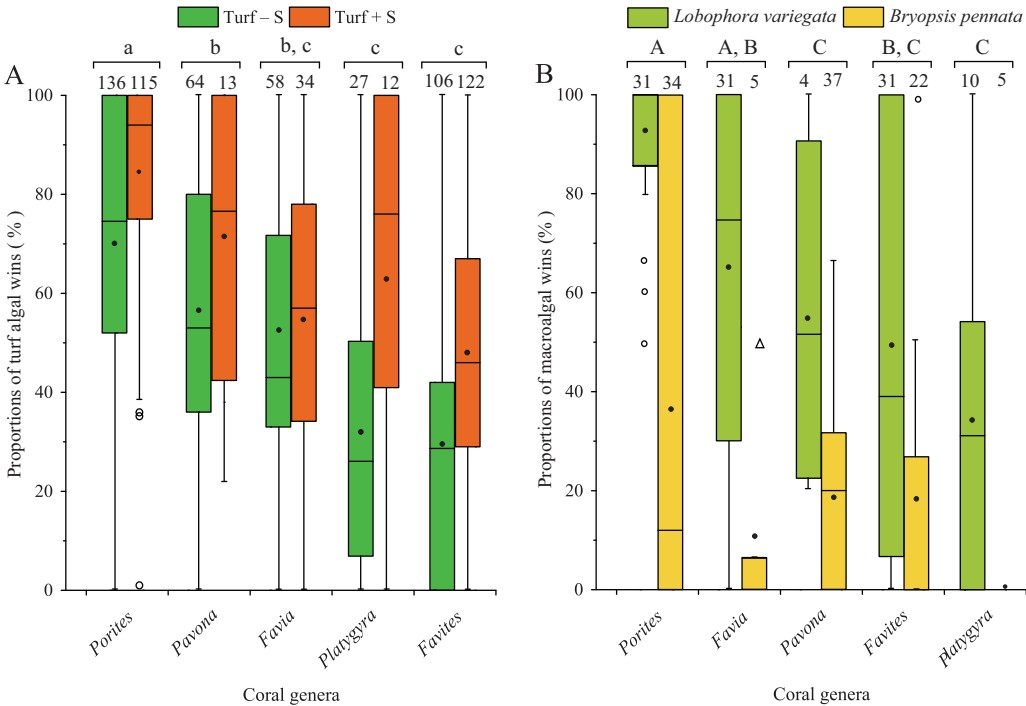

**Figure 6 Proportions of algal wins observed along edges of different coral genera.** (A) Proportions of turf algal wins on coral genera, with turf algae grouped into "Turf + S" (turf algae with sediments) and "Turf − S" (turf algae without sediments). (B) Proportions of macroalgal wins on coral genera. The numbers indicate the number of coral colonies included in the analyses. Similar letters (uppercase or lowercase) indicate significant differences (assessed by a two-way ANOVA) in post hoc comparisons for proportions of algal wins among coral genera (SNK test, $p < 0.05$).

(two-way ANOVA, macroalgae genera: $F = 40.48$, df $= 1, 181$, $p < 0.0001$; coral genera: $F = 9.35$, df $= 4, 181$, $p < 0.0001$; interaction: $F = 1.14$, df $= 4, 181$, $p = 0.3399$; Fig. 6B). Within the coral genera, algal wins of brown algae *L. variegata* were 2.5–5.7 times more frequent than those of green algae *B. pennata*. Both species caused the most harm to the *Porites* genus (93% ± 2.4% for *L. variegata* and 37% ± 7.6% for *B. pennata*; mean ± SE; Fig. 6B).

## Negative effects of turf algae on massive and encrusting corals

The effects of the coral growth form and the sediment on the outcomes of coral and turf algae interactions were analyzed for the *Porites*, *Favites*, *Favia*, and *Platygyra* genera. The proportions of algal wins varied significantly between the two types of turf algae, but did not vary significantly between the two coral growth forms (massive and encrusting), and a significant interaction was found between the types of turf algae and coral growth forms (two-way ANOVA, types of turf algae: $F = 28.42$, df $= 1, 585$, $p < 0.0001$; coral growth forms: $F = 1.26$, df $= 1, 585$, $p = 0.2625$; interaction: $F = 4.23$, df $= 1, 585$, $p = 0.0401$; Fig. 7). For the same coral growth forms, the proportions of turf algal wins increased significantly when sediments were trapped in the turf algae (Fig. 7). The proportions of algal wins of "Turf − S" were significantly higher on massive corals than

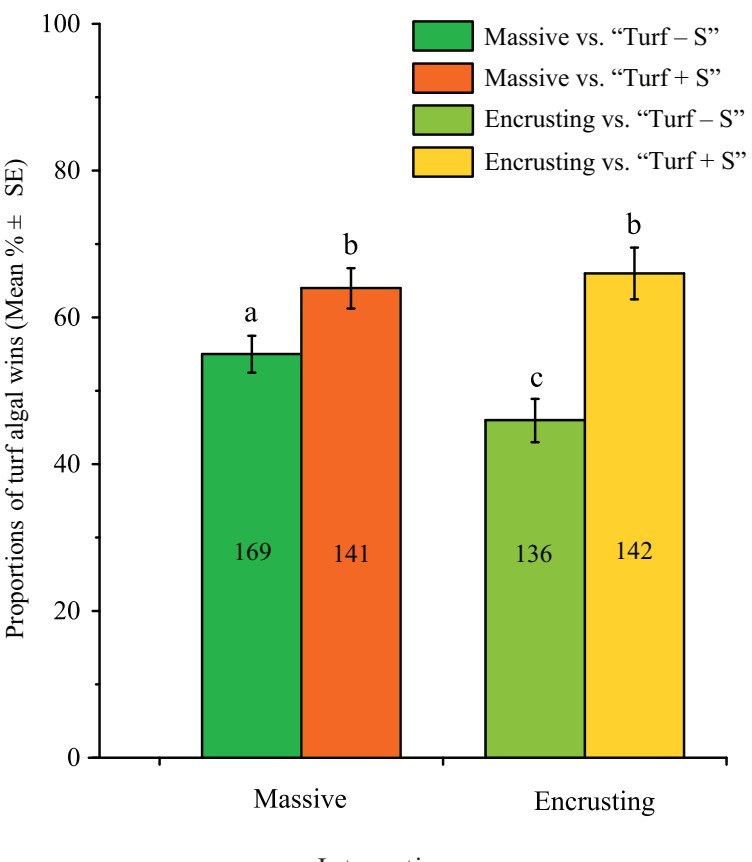

**Figure 7** **Proportions of turf algal wins observed along massive and encrusting coral edges.** Measured interaction groups were massive corals vs. "Turf − S", massive corals vs. "Turf + S", encrusting corals vs. "Turf − S", and encrusting corals vs. "Turf + S". "Turf + S" indicates turf algae with sediments and "Turf − S" indicates turf algae without sediments. The numbers indicate the number of coral colonies included in the analyses. Similar letters (uppercase or lowercase) indicate significant differences (assessed by a two-way ANOVA) in post hoc comparisons for proportions of algal wins among interaction groups (SNK test, $p < 0.05$).

on encrusting corals (Fig. 7). However, the proportions of algal wins caused by "Turf + S" did not differ significantly between these two coral growth forms (Fig. 7).

# DISCUSSION

## Coral-algal interactions varied with algal taxa

The majority of coral colonies competed with benthic algae, with interaction outcomes mostly being in favor of the algae rather than the corals. The interactions of the algae and corals varied among species and algal functional groups. Interactions between turf algae and corals were most frequent, with the proportions of algal wins and the proportion of algal contacts being considerably greater in turf algae than macroalgae (Table 1; Figs. 4 and 5). These findings have previously been observed on degraded coral reefs, where contact with turf algae resulted in algal overgrowth and caused bleaching or damage to adjacent corals (*Haas, El-Zibdah & Wild, 2010*; *Wild, Jantzen & Kremb, 2014*; *Swierts & Vermeij, 2016*). Some of our study findings were consistent with the results of these

previous studies, that is, corals were frequently observed to suffer overgrowth by turf algae; however, in the present study, coral bleaching was rarely observed. Although the causes of the observed algal overgrowth and coral bleaching were not surveyed in this study, previous studies have confirmed that turf algae can negatively influence corals by hypoxia and microbial growth (*Smith et al., 2006*; *Wangpraseurt et al., 2012*; *Gregg et al., 2013*; *Haas et al., 2013*; *Roach et al., 2017*). Turf algae may also influence corals via potential algal allelopathy (*Jompa & McCook, 2003*). Moreover, turf algae can rapidly increase in length and occur in both creeping and upright growth forms (*Hay, 1981*; *Connell, Foster & Airoldi, 2014*), which allow it to rapidly and densely overgrow adjacent corals.

Macroalgae caused relatively less damage than turf algae (Figs. 4B and 5B). Comparatively, *L. variegata* exhibited a greater damaging effect on corals than *B. pennata* (Fig. 6B). On degraded reefs, *L. variegata* is often widespread and competes with hard coral (*Hughes, 1994*). Because of its creeping growth pattern and thallus which can attach tightly to the coral surface, it is possible for *L. variegata* to smother and overgrow the subsurface of coral tissue (*McCook, Jompa & Diaz-Pulido, 2001*; *Nugues & Bak, 2006*; *Longo & Hay, 2015*). Meanwhile, bleaching on neighboring corals may be associated with allelopathic or microbial mechanisms during the overgrowth of *Lobophora* spp. (*Rasher & Hay, 2010*; *Vieira et al., 2016*). In contrast, *B. pennata* is characterized by a prostrate rhizoid and erect pinnule, which possibly account for its lower frequency of damage to corals in this study. The soft and short erect pinnules of *B. pennata* may have limited shading effects on adjacent corals. Other rhizophytic algae (e.g., *Caulerpa* spp.) in the Bryopsidales tend to bury their rhizoids in sand and sediment substrates for nutrient absorption (*Williams, 1984*; *Friedlander et al., 2006*), suggesting that hard coral may not be a suitable substrate for the growth of rhizophytic algae. Other investigators have reported that competition with *Bryopsis* sp. resulted in coral bleaching or tissue necrosis (*Barott et al., 2009*), but this was not recorded in our study. The effects of macroalgae on corals have been demonstrated, with a high variance in the potency of different algae and in the susceptibility of different corals (*Rasher & Hay, 2010*; *Rasher et al., 2011*). The interactions between macroalgae and corals varied among the different algal species, and these differences may be explained by algal growth patterns.

## Resistance to algal contact: driven by coral traits and growth forms

In our study, the proportions of algal wins of different algal interactions varied among coral genera (Fig. 6). We found that the *Porites* genus was more susceptible to damage by algal contact. Other studies have also demonstrated the sensitivity of *Porites* spp. to algal contact and showed that they often suffered algal overgrowth, tissue bleaching, or even mortality (*Titlyanov, Yakovleva & Titlyanova, 2007*; *Rasher & Hay, 2010*). The harmful influences of a specific macroalgae on corals may be associated with specific coral traits, including taxonomy, growth form, and colony and polyp size. *McCook, Jompa & Diaz-Pulido (2001)* hypothesized that size classes of the coral colonies and polyps may be associated with the ability of corals to compete against algae. It seems that corals with large polyps possess a higher tissue expansion potential than those with small polyps (*Erftemeijer et al., 2012*), which enables them to perform better in resisting foreign

matter invasion (e.g., algae, and sediments). This phenomenon may account for our observation of the *Porites* genus suffering greater damage from algal turf contact than the *Favites*, *Favia*, and *Platygyra* genera, all of which have larger polyp sizes.

Meanwhile, the coral growth form has also been found to be an important determinant of the outcomes of interactions between corals and turf algae (*Lirman, 2001*; *Haas, El-Zibdah & Wild, 2010*; *Swierts & Vermeij, 2016*). Previous research found that encrusting corals suffered the least harm from turf algae and that they had a higher percent of wins against turf algae than other coral growth forms (*Swierts & Vermeij, 2016*). The combination of sediments in turf algae appears to potentially alter the ability of the turf algae to damage different growth forms of coral (Fig. 7). The explanation for this may be that the combination of turf algae and sediments often exacerbates their damaging effects on adjacent corals (*Steneck, 1997*; *Erftemeijer et al., 2012*; *Goatley & Bellwood, 2013*). The proportions of algal wins of these combinations on massive and encrusting corals were extremely similar. In such cases, the morphology of colonies did not affect the coral-turf algal interactions. Thus, we hypothesized that the competitive advantage of the encrusting coral disappeared in the interactions between coral and turf algae when turf algae were combined with sediments.

## Sediments bound within turf algae can further affect the reef ecosystem

Our results suggested that the proportions of turf algal wins were augmented when sediments were trapped by turf algae (Figs. 6A and 7). Inshore coral reefs are usually exposed to high levels of suspended sediments (*Gilmour, 1999*), and turf algae can trap considerable quantities of sediments and reduce sediment resuspension (*Purcell, 2000*; *Gowan, Tootell & Carpenter, 2014*). Water flow is the key driver of sediments that influence the interactions between corals and turf algae (*Gowan, Tootell & Carpenter, 2014*). The water flow rate at Weizhou Island reef is low, with a seasonal range of 2.7–6.2 cm s$^{-1}$ (*Wei et al., 2017*). High sedimentation rates have also been recorded on the reef flat of Weizhou Island, with an average deposition rate of 2,157.9 g m$^{-2}$ d$^{-1}$ (*Wang, 2009*).

In addition, sediments in reefs contain an abundance of organic matter, and turf algae and sediment combinations have been shown to promote the growth of turf algae and inhibit grazing by herbivores (*Wilson & Bellwood, 1997*; *Bellwood & Fulton, 2008*; *Goatley & Bellwood, 2012*; *Goatley et al., 2016*). Herbivorous fish are a critical factor in coral reef environments, where grazing controls the abundance of benthic algae and coral-algal interactions (*Rasher & Hay, 2010*; *Bonaldo & Hay, 2014*; *Wild, Jantzen & Kremb, 2014*). The Reef Check on Weizhou Island reef showed that the average density of reef fish was only 0.03 ind m$^{-2}$ in the past few years (*Chen et al., 2016*), which is lower compared to that of the Luhuitou fringing reef (0.51 ind m$^{-2}$) 300 km from Weizhou Island (*Sun et al., 2018*). Thus, we speculated that the high frequency of coral-algal interactions and negative effects of turf algae on corals may be attributed to the increase in sediments trapped in the turf algae and the decrease in the abundance of herbivorous fish.

## CONCLUSIONS

The present study aimed to demonstrate the influence of species-specificity and sediment deposition on the outcomes of coral-algal interactions. We found that the outcomes of their interactions were related to biotic (i.e., taxa and growth patterns of algae, genus, polyp size, and coral growth forms) and abiotic factors (i.e., sediments). In the competition between corals and turf algae, sediments can exacerbate the ability of the turf algae to attack corals and weaken the ability of corals to resist algal invasion. Our data on the interactions between coral and algae were collected during a snapshot in time, and long-term field surveys and experiments need to be conducted to further understand the impact of sediments, their release of nutrients and housing of pathogens on coral-algal interactions as well as to determine the underlying mechanisms.

## ACKNOWLEDGEMENTS

Many thanks to Xiaoyan Chen and Zhenjun Qin for their constructive comments. We also thank Hainian Yu from the University of Queensland for English writing improvement. Many thanks to the captain Haichun Su. Finally, we would like to acknowledge, with great gratitude, the patience and support of two anonymous expert reviewers, whose detailed, constructive and highly encouraging feedback have helped us significantly in our refinement of this manuscript.

### Funding

This work was funded by the Guangxi scientific projects (Nos. AD17129063 and AA17204074), the National Science Foundation of China (No. 91428203), and the BaGui Fellowship of Guangxi Province (No. 2014BGXZGX03). The funders had no role in study design, data collection and analysis, decision to publish, or preparation of the manuscript.

### Grant Disclosures

The following grant information was disclosed by the authors:
Guangxi scientific projects: AD17129063 and AA17204074.
National Science Foundation of China: 91428203.
BaGui Fellowship of Guangxi Province: 2014BGXZGX03.

### Competing Interests

The authors declare that they have no competing interests

### Author Contributions

- Zhiheng Liao conceived and designed the experiments, performed the experiments, analyzed the data, prepared figures and/or tables, authored or reviewed drafts of the paper, approved the final draft.
- Kefu Yu conceived and designed the experiments, contributed reagents/materials/ analysis tools, authored or reviewed drafts of the paper, approved the final draft.

- Yinghui Wang contributed reagents/materials/analysis tools, approved the final draft.
- Xueyong Huang performed the experiments, approved the final draft.
- Lijia Xu authored or reviewed drafts of the paper, approved the final draft.

## Field Study Permissions

The following information was supplied relating to field study approvals (i.e., approving body and any reference numbers):

Field research was permitted by the School of Marine Sciences, Guangxi University.

## Data Availability

The raw data are provided in a Supplemental File.

## Supplemental Information

Supplemental information for this article can be found online at http://dx.doi.org/10.7717/peerj.6590#supplemental-information.

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
