# Peer review of "Coral-algal interactions at Weizhou Island in the northern South China Sea: variations by taxa and the exacerbating impact of sediments trapped in turf algae"

_PeerJ, doi:10.7717/peerj.6590_

## Round 0.1 · original submission · Major Revisions

Your manuscript has now been reviewed by two experts in the field, both with extensive experience studying coral-algal interactions. Both felt that the work was of interest and the experiments worth publishing, but both had many many recommendations for improvement or areas where the work was unclear.

Their comments are wide ranging and extensive, addressing many aspects of the research. All of their recommendations are in line with expectations of basic reporting quality for PeerJ, with a number of recommendations on improvements to statistical analyses, data visualizations in figures, and interpretations of results. They also were both very generous with their efforts to improve the language in the manuscript, and both recommended that you make every effort to have the English improved by a native speaker before submission.

Given the extensive edits recommended by the reviewers, I expect that your revision will address each recommendation point by point; if you disagree with a recommendation you must explain your rationale and justify your decision to not follow their advice. I expect that this manuscript will be reviewed again and, as noted, there is no guarantee that it will be accepted. However, I do believe that the manuscript will be vastly improved by the excellent recommendations of these expert reviewers. Please acknowledge them in your acknowledgements section.

Best of luck with the revisions, and we look forward to your revised draft.

Reviewer 1 ·

Basic reporting

Throughout most of the manuscript, the English language needs improvement. For example, lines 26-27, 29-31, 39-42 could be improved because the current phrasing is difficult to understand. In the line by line comments I stated “unclear” to indicate some of the hardest to understand

Also, it should be Lobophora variegata not variegate

Several of the studies that were cited were not the best citations for the statements made and were often misleading. For example:

Lines 237-8 Neither of those studies looked at allelopathy in turfs – the Rasher and Hay only tested allelopathic effects of macroalgae and the Jompa and McCook study hypothesized that effects of the turf species were allelopathic.

References to McCook 2001’s review paper- polyp and tentacle and coral sex were not the focus of any of these studies were hypothesized, but not tested, please make it clear in the text when citations are based on ideas from other papers and have not been explicitly tested

Line 291: Carpenter and Williams 1993 is not about sediment, it was about algal turf height due to grazing and flow profiles, and the implications for reef metabolism, a reference in regard to sediment seems out of place.

The structure of the article was clear.

The hypotheses should be explicitly stated in the introduction, they were mostly relayed in the methods and the results.

From the goals, especially for the sediment-algal turf surveys, I expected a manipulative experiment not a survey- this should be clear in the introduction.

Experimental design

The aspects of this research that are addressing research gaps needs to be emphasized to help focus readers in the introduction

More details are needed in the methods section in order to ensure that this project could be replicated. Below are examples of the parts that were difficult to understand

There should be information about the study site and the corals and algae that are being studied, especially since so much detail was completed on differences among coral functional groups. Massive and encrusting should be defined and which corals belong to those categories should be stated explicitly. Additionally, ~50% of the benthic composition were corals and algae, but what made up the remaining 50% ?

Different sites were examined, but there is no information about why there are different sites and what the differences among the sites are.

Additionally, how far apart were the transects, were they meant to describe these sites?

Another aspect of the experimental design that was not clear was if the same transects/quadrats were used for all of the sub studies (coral cover, macroalgal cover, coral-algal interaction extent)

More information about the video transects would be helpful for the repeatability of this project. Additionally, the pixels of the camera would be helpful to get an idea of how clear the image was for identification. Here are some of the questions I have”
What was the distance between the camera and the benthos?
Were the distances consistent?
How were the videos completed, with a diver?
What happens if a coral colony extends across the distances sampled?
How does that affect the coral extent measurements?

Also – how were distances along the transect determined to make coral cover measurements – were they random?

It was not clear how the authors determined colonies – what defined a colony? Could the same colony extend past a quadrat?

Top-view photos were not explained before they were mentioned or how that was achieved

The descriptions of extent of algal contact and how percentage of contact were determined should be clarified
i.e., was it the percentage for a particular colony or for all of the colonies in a quadrat?

Only algal winners were described, coral wins and neutral interactions need to be described as well to help understand how these categories are assessed

To determine sediment cover it was not clear from the description or from figure 2 how sediment cover was determined if the turf was not heavily covered in sediment – even the no sediment seemed to have some sediment. Were photos zoomed in to determine this? By how much?

In the statistical designs, careful attention should be taken to the pseudoreplication of this project. From the descriptions of the methods and statistics it appears that all quadrats or all points or colonies are treated independently, please see Hurlbert 1989 for more information about pseudoreplication.

Validity of the findings

Based on the statements made in the results, some of the statistical tests used were not the best to test for the interactions of interest. This is particularly true for the ANOVAs that were run for morphology vs sediment presence/absence. It was treated as a one-way ANOVA, but I think it would be better treated as a 2-way ANOVA with morphology and turf sediment, this way, if there is an interaction between morphology and sediment (which was suggested based on conclusions made about encrusting corals and the presence of sediments), could be made with statistics backing it up. Here is also an instance where pseudoreplication could be taken care of – i.e., if colony was nested within transect (although this is still ignoring site).

Several statements in the discussion were stronger than they should be based on the data collected and other studies.
For example, line 232 states that turf “always results in the bleaching, overgrowing or killing of corals,” which was not true in the study that was conducted here or in the studies that were cited.

Additionally, the first paragraph discusses bleaching vs overgrowth of turfs, which was not directly tested in this study.

Additional comments

Additional sources for sedimentation and coral-algal interactions

McCook, L. (2001). Competition between corals and algal turfs along a gradient of terrestrial influence in the nearshore central Great Barrier Reef. Coral Reefs, 19(4), 419-425.

Gowan, J.C., Tootell, J.S. & Carpenter, R.C. Coral Reefs (2014) 33: 651. https://doi.org/10.1007/s00338-014-1154-1


Figures:

Figure 2: It is not clear what the description of 2D means. From the figure legend it sounds as though the sediment was manipulated, however, based on the text, it does not seem as though sediment presence/absence was manipulated.

Raw data:
Please provide metadata for each of the datasets (i.e., sheets).
Benthic cover sheet
Live coral cover (m) is this referring to the point where the measurement was made? How were these locations chosen?
Why are there fewer points for W4-2?

In Algal composition around coral sheet
“No algal contact” would be good to show in the data, because otherwise it seems as though every coral is in contact with algae.
Misspelling: coral border

On the following sheets: algal composition, competition outcomes and turfs effect on corals there is no information about the sites and transects.

Line by line comments (some repeated from overall comments)

Abstract

Lines 26-27: wording unclear, should be:
Comparing coral-algal interactions in the presence and absence of sediment showed a significantly more negative effect of turf algae on massive and encrusting corals

Lines 29-31: wording unclear

Introduction

Lines 39-42: wording unclear

Line 48 – polyp and tentacle and coral sex were not the focus of any of these studies , and in some cases (i.e., McCook 2001) were hypothesized, but not tested, please make it clear in the text that these are hypothesized, but not tested.
Line 54 – references have two Barott 2012 papers – be clear which one this is
Line 69 – damaged contiguous tissue wording is unclear
Line 77 – “later trap sediments” not clear
Line 92 –Lobophora variegata not variegate?


Methods

Line 117-118 – were these transects in the same location as the macroalgal surveys or were the methods just the same? The wording for these sentences was unclear.

Line 121 – within the quadrats? I think more information is needed about the video transects – what was the distance between the camera and the benthos? Was a quadrat videoed for the extent? What if the same coral colony extended across more than one quadrat?

Line 124 –how did you a define a colony in a quadrat?

Line 128 – first time a ruler is mentioned and top view pictures – not sure what this is referring to, needs a clearer description of how the extent was measured and how percentage determined – the total perimeter for that coral or within the quadrat determined the percentage of contact?

Line 131 – I do not understand this description for how outcomes of interactions were identified

Line 132 – for each colony, how was win/loss determined? For each algal group associated with the coral, or once per colony in a quadrat? Were multiple outcomes assessed for a single colony?

Line 133 – how were coral wins and neutral interactions identified?

Line 140 – this was all identified from the photos? How magnified were the photos to assess sediment cover? Was there variation in the amount of sediment cover? Again, was this 1 interaction per colony or for every measurement with an extent?
From the introduction I expected a manipulation here, not a survey of sediment no sediment.

Line 148 – date = data?
Line 149 – wording unclear

Line 150 – does “on the same colony” mean for each individual coral or across coral genera – I think it means individual coral from your photos and raw data, but that should be clarified.


Results

Line 169 – The major algal functional group was turf algae (24%). 

Line 211 – I think t-tests were completed, but what about a 2 way ANOVA comparing coral genus and algal type? That way the statements can be made for determining which corals experienced more damage and which algae created more damage in a single analysis

Line 218 - How were letters determined? Through a post hoc test? Which one?

Line 186 – varied significantly?

Lines 207-8, unclear wording

Line 232 – “always result in the bleaching or overgrowth…” of corals is too strong of a statement and the data from this study does not support that conclusion.

Line 234 – instead of “confirmed”, “supported” would be a better word

Line 235-6 – unclear wording – remove "the" before bleaching

Lines 237-8 neither of those studies looked at allelopathy in turfs – the Rasher and Hay study only tested allelopathy of macroalgae and the Jompa and McCook study hypothesized that effects were allelopathic

Line 240 – unclear wording

Line 241 – none of the data shows a bleaching effect of algal turf – this should be clear that this is an observation

Line 242- direct overgrowth

Lines 260-264 – unclear wording “at present research”; “Porites has been found…, yet…” - I think should be an “and” instead of “yet”; “more dependent on the specificity of the corals” not sure what this is supposed to mean

Line 265 – again, I think that polyp size was a discussion point made by McCook and not something that has been tested. If the present authors want to test that hypothesis, they should do so explicitly, which was not completed, although it was discussed as though it was (line 271)

Line 291 Carpenter and Williams not about sediment, it was about algal turf height due to grazing and flow profiles, and the implications for reef metabolism reference in regard to sediment seems out of place.

Lines 304-5, unclear language

Conclusion

Line 307 – too strong of a statement that turf outcompetes corals more than macroalgae and claiming that it “confirms” this, which I do not think is well established. Additionally, because the opportunity for coral-algal contacts between algal turf and coral were generally higher than between coral and macroalgae it is hard to state that algal turf is a better competitor with corals than macroalgae, especially if other interactions with macroalgae maybe led to coral death, thus there would be no evidence of the contact on the reef.

Too much emphasis on large and small polyps without directly testing it – this is a result that has been discussed (i.e., in McCook 2001), but not explicitly tested

Reviewer 2 ·

Basic reporting

Basic Reporting:

Clear unambiguous professional English language used throughout: NO.

The English grammar and writing need some work. There are numerous grammatical mistakes throughout the paper. There are also many sentences which are unclear. To highlight these issues, I refer to the last two sentences in the abstract (Lines 26-31). The use of presence and present is incorrect (Line 27), and there is incorrect spacing between words (Line 28). The last sentence of the abstract begins with a dangling participle phrase, which makes the sentence very unclear. There are other numerous grammatical errors throughout, which make the paper difficult to understand, including changing tense within paragraphs and even within the same sentence (see for example lines 185-186) and numerous run-on sentences. There are also many uses of the noun form of words as adjectives (e.g., line 205). Lastly, the authors switch freely between British and American spelling of words.

Intro and background to show context. Literature well-referenced &relevant: OK

The introduction, as it stands, is relatively well-referenced. However, it is a bit of a collection of facts rather than a logically organized introduction to the material that clearly discusses current knowledge gaps and problem in the field that the authors’ work serves to address.

Structure conforms to PeerJ standards, discipline norm, or improved for clarity: OK

Most of the structure and formatting of the paper seems to conform to the standard norms in the field. One small note is that the in-text references do not have a comma between author and the date. Also, in the reference list, the DOI numbers should be directly following the end of the reference rather than being on a separate line with a different indention. Lastly, as per PeerJ standards, there should not be funding sources included in the acknowledgments section.

Figures are relevant, high quality, well-labeled and described: NO

Figure 1: Countries other than China should be labeled on the map. Latitude and longitude, as well as a scale bar, should be included in the top panel. A table should also be included that details the site level data such as latitude, longitude, coral cover, and the number of samples taken at each site. Much of this aforementioned metadata could also be shown visually on the map.

Figure 2: Apart from the poorly written figure caption, this figure is mostly OK. I would suggest a scale bar in panel E. Also, after looking at the other panels it is concerning that most of their corals look incredibly small in these interactions (i.e., < 5 cm in diameter).
Figure 3: For all of the bar graphs, I suggest that they change them to box and whisker plots and include the number of samples (i.e., n = ?). The Genus and species names should be italicized.

Figure 4: This needs to be a box and whisker plot. There should also be some indication of significant differences between groups. Both panel A and B should have the horizontal axis labeled. Panel B has no indication of the variance. There needs to be an indication of sample numbers for each group (i.e., n = ?). All of the same goes for Figure 5. Also, there should be consistency in the use of the words corals edge. Throughout the paper, it is written as “corals edge” and “coral’s edge”, when I think they actually mean corals’ edge.

Figure 6: What happened to the Lobophora bar for Pavona, and the Bryopsis bar for Platyyra. Once again, these would be much more effective as box and whisker plots.

Figure 7: The +S and -S should be defined in the caption, i.e., they should not say that “+S = Turfs + S” they should say that +S = Turfs with sediment.

Table 1: they should report their actual p-values, not just *s. They should also report their sample sizes.

Table 2: They need a more descriptive caption. Again, report the actual p-value.
Raw data supplied: Yes

Experimental design

Original primary research within the scope of the journal. Yes

Methods described with sufficient detail and information to replicate. NO
In some parts of the methods, it was hard for me to follow due to grammar errors and poor clarity. In the Statistical analyses section, the authors say that “date [I assume they meant data] were transformed”, but they do not explain how.

Validity of the findings

Validity of finding

Data is robust statistically sound and controlled. Not really

The data in the results section is presented like there is just a single number for percent of the perimeter in contact with algae, types of algae in contact with coral, etc. Are these really single values, e.g., was there actually 44% of all coral perimeters in total that came into contact with algae (line 174) or was there on average 44% of each coral colony that was in contact with algae? It seems that the authors should report the average percent perimeter in contact with algae, not just the overall total. If these numbers are averages then they should report the sample size for each group, the mean, and the standard deviation, i.e., (n = x, mean + std). Also, throughout the results section and in the tables the authors should report the actual p-values and not just p<X such as in line lines 180, 182 and elsewhere. The authors should also report p-values when they state that things were not significant, or that results were similar between groups.

They should also make sure that proper and consistent use of the words “between” and “among” is used throughout the paper, as these words imply two different things statistically. For example, in line 185-186 the authors state that “turf algae was the largest and vary [grammar error] significantly among coral genera…” However, it seems as though they actually mean that this varied significantly between genera not among them. Another example is line 216-217 “negative effects…differ from coral genera.” This should also be between coral genera.

In the Statistical analyses section, the authors state that they transformed the data to be homoscedastic and then performed ANOVAs, but if the data still was not homoscedastic then they used non-parametric analyses. Why not just use non-parametric analyses to begin with instead of transforming the data and then using different tests for different portions of the dataset.

Conclusions are well stated, linked to the original research question, and limited to supporting results. NO

The discussion section makes many conclusions that are not well-supported by the data. Some examples include the following:
1) Lines 255-257: “The interactions between macroalgae and coral varied with algae species, and these differences may be ascribe to the growth pattern and the ability of allelopathic of algae.” Apart from the numerous grammar errors in this sentence, the authors also provide no data which would establish a role of allelopathy in the effects of algae on coral.
2) Lines 301-304: “Hence we conclude that the high frequency of coral–algae contacts and the more negative effects of turf algae on corals should be attributed to the increasing of sediment trapped in turf algae and the decreasing abundance of herbivorous fish.” Here the authors conclude that herbivorous fish somehow play a role in the effects of algae on coral. Although it is plausible that fish may play some role, the authors present no data on fish and do not even provide references from the literature that might support this claim.
3) Lines 231-232: “ Turf algae… can always result in bleaching…” Here the authors discuss the role of turf algae in bleaching although they do not measure bleaching at all in the study. Also, always is a major overstatement.
Other examples include lines 241-242, lines 246-248, and lines 281-284.

Additional comments

This manuscript by Liao et al. provide in situ data from the South China Sea to argue that the frequency, extent, and outcome of corals interacting with benthic macroalgae are a function of the type of species of the coral. They also argue that the settlement of sediment in turf algal assemblages enhances the negative effects of algae on coral.
The manuscript is within the scope of the journal and would be of general interest to coral reef biologists. Although the manuscript follows the formatting guidelines of PeerJ, it suffers greatly from a lack of correct English grammar and sentence structure. This leads to the manuscript being hard to follow and confusing at some points. Aside from the grammar and English the visualization of the data, and overall conclusions need some improvements as listed below. As the manuscript stands currently, it is not fit for publication in PeerJ. I would suggest to reject the manuscript with a chance for resubmission after major revisions.

---

## Round 0.2 · Major Revisions

Your revised manuscript has now been reviewed by two experts in the field, both with extensive experience studying coral-algal interactions. Their assessment of your revised manuscript included a number of recommendations for improvement, and as editor I am requesting that you submit a revised version of the manuscript that addresses each of their points, along with documentation of the changes made to address each point or a rebuttal of each point.

However, both reviewers expressed considerable frustration at the English grammar, which was sufficiently poor to prevent them from assessing the scientific validity (methods, results, conclusions and interpretations) of the work. I must be firm that if the authors do not seek professional copy editing and a thorough grammatical revision from a native english speaker I will be forced to reject the manuscript from publication, as it is crucial that scientific publications be clearly understood and this requires proper English grammar (for better or worse).

Reviewer 1 ·

Basic reporting

I struggled with English grammar, word choice, phrasing and unclear definitions throughout the manuscript. This continues to make understanding the manuscript difficult. Additionally, there were several instances of misspellings or poor proofreading (e.g., missing the end of the word, or the incorrect word), including spelling errors that I had previously commented on (e.g., Lobophora variegata not variegate). In terms of unclear definitions it was often not clear what was described, for example – line 55-57, it was not clear to me what dynamics were being referred to, nor what was meant by the algal properties. Other instances were the use of negative effects of algae – was this the frequency of algal winners?
Intro & background to show context. Literature well referenced & relevant.
I found the intro and background to be much improved, but the hypotheses/objectives were not clear, in particular the third objective. I did not understand the motivation or what was referred to in terms of “morphological advantage”, I think some of the information in the discussion could be used to motivate this point, in particular line 318.

Structure conforms to PeerJ standards, discipline norm, or improved for clarity.
Yes, this was fine

Figures are relevant, high quality, well labelled & described.
Figure 1 – no error associated with the means, no information about transects
Figure 2 – labeling should be more specific to each photo. 2D – it’s not clear if the sediments were physically removed by the researcher or if the photo shows low sediment
Figures 4 and 5 – I do not understand the difference between the purple and green shading in the bars. Why there are capital and lower case letters for the SNK post hoc tests is not clear
Figure 6 – SNK results not clear based on what is written in the figure caption –SNK seems to shows the differences among the genera, not when there are differences within a genera.
Figure 7 – what is “negative effect” is it the same as an algal win? This was never defined…This result was confusing because the asterisk use makes the results seem like a t-test was used (which was written in the text), but an interaction result was shown, indicating that a two-way ANOVA was run on the data, and a multiple comparisons test (like SNK) should be used, as a t-test will increase your probability of type I error.

Table – Unclear what the p values associated with each genera (the last column), or the last row (Between genera) was referring.

Raw data supplied (see PeerJ policy).
Summarized data provided (Table 1)

Experimental design

EXPERIMENTAL DESIGN
Original primary research within Scope of the journal.
Research question well defined, relevant & meaningful. It is stated how the research fills an identified knowledge gap.
This is improved upon revision, but as above, I found the objectives of the study not stated clearly, and the points highlighted here (e.g., meaningfulness), not stated clearly.
Rigorous investigation performed to a high technical & ethical standard.
Because I had difficulty understanding the methods and analyses, I cannot assess this. Additionally, the results do not clearly show the results of the tests used, nor which test was used – whether it was an ANOVA or a Kruskal-Wallis test.
Methods described with sufficient detail & information to replicate.
I struggled with the methods. In particular the use of length, is this meant to indicate perimeter of corals? Was it measured from the photographs? Was anything measured in the field? How was the line-intercept method employed versus CPCe?
Speculation is welcome, but should be identified as such.
There were several instances in the discussion (see line by line comments) where there were statements made with no clear data to back them up
Conclusions are well stated, linked to original research question & limited to supporting results.
Again, there was the use of dynamic, which was not supported by the results. The last statement is a strong statement, especially in regard to their last result – where it’s the an interaction between sediment and coral growth form that highlights differences in (I think Algal wins?) as the severity of the effects of sediment depend on coral growth form.
The discussion over states the results, as there were points made that were not explicitly compared (e.g., polyp size of corals, “damage” effects of algae).

Validity of the findings

VALIDITY OF THE FINDINGS
Impact and novelty not assessed. Negative/inconclusive results accepted. Meaningful replication encouraged where rationale & benefit to literature is clearly stated.
Data is robust, statistically sound, & controlled.
Although there was discussion of sites and transects, there was no description of how transects or sites were taken into account in data analysis. For example, were results averaged over quadrat or transect? What is the level of replication?
I found parts of the statistics section difficult to understand. In particular, I did not understand why a t-test was run after conducting a two-way ANOVA. This is inappropriate, especially given the significant interaction between genera and presence of sediment.
Additionally, although p-values were given, there were no degrees of freedom or F-values provided to aid in the replication of the results (and give an idea of replication).
For the results, such as in line 220, information is provided for overgrowth and bleaching, but no data is shown to compare these (unless green and purple are supposed to indicate that?)
In the methods, results and discussion, there were several misleading statements. For example “negative effects of algae on corals” was often made, what is this referring to? It sounds as though an experiment was conducted or a long-term survey. I think it may be referring to algal wins. However, as I read this, this survey happened only once, and so this shows a snapshot in time, not necessarily long-term effects of algae on corals. Thus these results are not “dynamic” which is another point that was made by the authors (e.g., line 352). There should be a discussion point about how the data taken were from a snapshot in time, and not necessarily indicative of dynamics of coral-algal interactions on reefs.
Line 263 – I found the interpretation of the interaction difficult to understand, and really do not understand why a t-test was conducted after the two way ANOVA and not a post hoc test that would take into account multiple comparisons.

Additional comments

Line by Line comments:
Abstract –
Line 24: “Absent the effects of sediments” – is poor workding\
Line 29: “Morphological superiority” – unclear what this means
Line 42-44: poor English
Line 53: “witch” should be which
Line 55-58: not clear
Line 71: Logic bounces back and forth between turfs having an effect and not having an effect, which is confusing
Line 75: Unclear what “cause in underlying benthic communities” refers to
Line 77: Not clear how water flow influences sediments in this sentence, or why this is relevant
Line 84-88: I think this is meant to help define the knowledge gap, but does not highlight how this work fills in a meaningful gap in knowledge – but the critique that it starts with (“field surveys of coral-algal interactions are still insufficient….) is exactly what the authors do in this study
Line 97-98: Again, “morphological superiority” is not clearly defined, so this hypothesis is not clear.

Methods
Line 118 – runon sentence
Line 122-23: I do not understand how CPCe vs the line intercept method was used. Why was coral length measured? Is this supposed to be the perimeter of each coral? Was this done in the field?
Line 125: m/min? not m/mine?
Line 132-134: unclear, not sure what this means
Line 146: unclear wording “interfacial algae” and “damaging adjoin algae”
Lines 152-154: unclear what growth form means – is this meant to be analogous to morphology? Does upright mean branching?
Line 158: “large” should be “larger”
Line 162: poor grammar
Line 179 – not clear if negative effects indicates algal wins or a different designation was used.
Line 181- the rationale for the t-test is not clear, and it is an inappropriate test to use, why not an SNK test?

Results
Line 198 – the number of corals and total perimeter: does this mean that site and transect were ignored? Were there corals with multiple measurements for proportions of wins/losses/neutral? If so, then a treating coral, at least, as random effect would be appropriate (because it’s possible that there is variation due to the specific corals used).
Line 235: unclear grammar

Line 250: “damaging effects” is not clear, again, is this algal wins? This statement makes it sound like a longer –term survey or experiment was conducted
Line 256: “to” should be “on”
Line 263: unclear grammar
Discussion:
Line 278: no data to support this statement
Line 282: “allelochemical competitions” not clear grammar
Line 284-5: high density of biomass – does not make sense
Line 302: “may be ascribe…” – not clear grammar
Line 314: first time since the introduction that polyp size was introduced – not data on this
Line 334: use “comparatively” but there are no comparisons
Line 337: units incorrect, I think “a” should be “s”
Line 352: “dynamics” were not studied

Reviewer 2 ·

Basic reporting

Clear unambiguous professional English language used throughout: NO
Although many of the grammatical mistakes in the first version have been corrected, there are still numerous grammar and spelling errors throughout the paper. This makes the paper hard to read and difficult to understand in some areas.

To highlight the grammar errors I refer to the abstract and introduction.

Line 21: The word "with" should be replaced by the word "between".
Line 24-25: The sentence which reads, "Absent the effects of sediments, the negative effects of turf algae on coral genera increased 1.1-1.9 times." does not make sense. The following sentence is also confusing, as it starts out by talking about comparing the presence and absence of sediment, but then concludes by talking about the effects of coral morphology.
Line 29: "competitive" should be "competing"
Line 42 -44: "The outcomes of such interactions are mainly determined by the species involved, since the variability of mechanisms and strength of different coral-algal interactions." is a sentence fragment, not a complete sentence.
Line 47-50: The phrase beginning with "Barott et al." on line 47 is a run-on sentence.
LIne55-58: another run-on sentence
Line 68: "damaged" should be "damage"
Line 70: Subject verb agreement: "seems" should be "seem"
Line 84: "researches" should be "researchers"
Line 85: "interaction" should be plural
Line 97 is missing a preposition. it should probably read, "... enable their competitive overgrowth of coral..."

This list of errors is by no means comprehensive, and is only what I found in a quick pass of the abstract and intro, but the grammar and English errors don't stop there, as there are still numerous errors throughout the entire paper.

Introduction and background show context, literature is well-referenced and relevant: OK

The introduction has definitely improved, however, there are still some places where statements need to be supported by a reference. Some examples include the following:

Line 35-36: "This is particularly pronounced in reefs where human activity is frequent. " should have a supporting reference

Line 36-39: The sentence starting on line 36, which reads, "...owing to overfishing (REF), eutrophication (REF), sediment deposition (REF), and global climate change (REF)." should have a reference for each of the proposed factors.

Line 42: The statement, "The outcome of such interactions are mainly determined by the species involved..." needs a reference

structure conforms to PeerJ standards: Yes

Experimental design

Methods described with sufficient detail and information to replicate: OK

The numerous grammar errors in this section still make it a little hard to follow.

My biggest concern was the authors' statistical analyses. They state that if data were not homoscedastic then they log transformed the data and used parametric statistics. In their rebuttal they state that they did this because they " think that the accuracy of the parametric test is higher than the non-parametric test.", but this is simply not true. It is not about being "more accurate" it is about using the appropriate statistical test for the dataset. By log transforming the data they do meet homoscedasticity but this is only because they are not only logrithmically compressing the data, but also logrithmically compressing the variance. Thus, when they log transform the data and then use paramtric tests they are more likely to get significant p-values. They even state in their rebuttal that they know that the non-parametric tests on the untransformed data may report non-significant p-values, while the parametric tests may report significance. To me this basically seems like a form of p-hacking. If the data do not meet the requirements of a statistical test, don't transform the data until they meet the requirements; instead use the appropriate statistical test for the type of data you have. This would also solve the problem of having different analyses used for figures 4 and 5.

Validity of the findings

Figures are relevant, high quality, well-labeled and describe: NO

Figure 1 is much better!

Figure 2 is fine.

Figure 3: The caption should read, "percent of coral on coral reef of Weizhou Island", not "in". Also, the caption should be a bit more descriptive. Lastly, the Figure 3 panel B reports data for B. Plumosa, but in the text they report on B. pennata, and in Figure 6B they report data for B. plumose. This is a blatant mistake and needs to be corrected.

Figure 4: I have read the author rebuttal, but still think that at least panel A should be shown as a box and whisker plot so that the entire spread of the data can be seen. Also, why are both purple and green used to indicate algae winning, is there a difference between the purple win, and the green win? Lastly, there should be a legend for the colors in the figure not just a description of the colors in the caption.

Figure 5: Same as Figure 4 Also, I don't like that they used different analyses for the two different figures (i.e., parametric tests in Figure 5 and non-parametric tests in Figure 4).

Figure 6: once again I still think that box and whisker plots would be more informative so that readers can see the full variance in the data. Also, I think that both vertical axes should be labeled in panels A and B.

Figure 7: "Asterisk" should be plural (i.e., asterisks). Also, significant differences should also be shown between the two growth forms.

Raw data supplied: Yes

Original primary research within the scope of the journal: Yes

Conclusions are well-stated, linked to the original research question, and limited to supporting results: OK

The discussion and conclusion have significantly improved from the first submission, but I still have a couple of small comments, which follow.

There are still some grammar and spelling mistakes in this section

Line 281: They might want to add the references Microbial bioenergetics of coral-algal interactions (Roach et al, 2017), Visualization of oxygen distribution patterns caused by coral and algae (Haas et al, 2103), and Biological oxygen demand optode analysis of coral reef-associated microbial communities exposed to algal exudates (Gregg et al, 2013) to the references about microbial growth and hypoxia, as these recent publications show a clear link between microbial growth and algal induced hypoxia.

Line 337: "a¬-1" should probably be "s-1", as it is talking about a rate.

Line 344-345: The authors state that the fish density of 0.03 ind m2 at Wheizhou reef is "high compared to Luhuitou fringing reef (0.51 ind m2)", but it is clear that the fish density is actually lower at Wheizhou than at Luhuitou.

Additional comments

Overall, the revised manuscript by Liao et al. has made some major improvements since its first submission. However, the manuscript still suffers from poor grammar and English, which make it hard to read, and will necessitate further revisions. The manuscript in its current form is still not acceptable for publication, but could probably be published with what are mostly minor revisions.

---

## Round 0.3 · Major Revisions

Both reviewers have gone to considerable lengths to provide a final round of edits. Their efforts are very much appreciated and I encourage you to address all of their additional recommendations in a revised version. They both agreed that the English grammar was greatly improved, and have made additional recommendations in that area.

Reviewer 1 ·

Basic reporting

Fine

Experimental design

See specific comments

Validity of the findings

I found the statistical design and the presentation in the results not completely clear, two tests appear to be used to evaluate the extent of interaction, one a parametric and one a non-parametric test. This needs to be clarified before publication.

Additionally, figure 2d contains a photo where sediments were removed by "blow air", but the methods do not mention a manipulative aspect to this study. This needs to be clarified and explained before publication.

Please see the specific comments

Additional comments

In general, I find this manuscript much improved. However, I am not sure of the comparisons that are conducted across genera for the extents of coral-algal interactions. It is not clear from the methods and results which tests are comparing across all coral genera, the total linear extent of algae; or specific comparisons of the extent of turf, macroalgae, cca, other. The table does not help to clarify this point. Additionally, based on figure 2 – there seems to be a manipulative aspect to this manuscript (removal of sediments) that is not explained in the methods, or the figure does not accurately represent the methods.



Abstract: fine

Introduction:

Line 86-9: I think there is something missing here – Based on this statement, I would not expect this paper to be based on surveys. Maybe the authors mean that field surveys often do not take into account the diversity of traits associated with corals and the environment that could influence the outcome of coral-algal interactions?


Methods:

line 132: how were corals chosen?
line 156: what growth forms?, ah, it’s below, move up here
line 163: Specifiy which corals (by genera) are characterized as small and large polyps


I was a bit lost in the analyses that were completed here for similar comparisons, and table 1 did not help me understand what analyses go with what comparison. Maybe referencing the hypothesis in regard to the way in which the data were analyzed. Additionally, I do not understand why a two way ANOVA was not used cross algal type and coral genus, instead of a Kruskal-Wallis and a one way ANOVA.

It would be helpful to clarify in the figures what test was done to determine significant differences, and if the results of multiple tests are on the same figure (e.g., multiple 1 way ANOVAs or Kruskal-Wallis test if each coral genus was assessed separately).

Differences in the proportions of different algal contacts (with either turf algae and
176 macroalgae) among coral genera were assessed using a Kruskal-Wallis test.

line 177: what does “Algal types” mean here?
A one-way ANOVA was used to compare the proportions of different algal types on the coral edges within a coral genus.

line 178: what does specific coral-algal competitive outcomes means? Is this referring to the wins, neutral and loss, or between specific coral-algal generas?

To assess the differences in
181 proportions of algal wins, a two-way ANOVA (with turf algae type/macroalgae genus and coral
182 genus as fixed factors) was used, followed by a SNK test.

To assess the differences in the
183 proportions of algal wins and the similarity among groupings, a two-way ANOVA (with turf
184 algae type and coral growth form as fixed factors) was used, followed by a SNK test.

Results are
185 presented as mean =standard error (SE).
line 183 – similarity among groupings is not clear


line 185 – most data are shown as boxplots, why does it say mean +/- SE?

line 197- 208– Now I am confused about what data was used for the Kruskal-Wallis and 1-way ANOVA, based on the summary of the results it sounds like they were testing the same thing (Coral Genus and total extent of coral algal interactions).

line 207 – remove “was”
line 238 – turf algae types? Is this with or without sediment?

Discussion
Are there size differences across these corals? Are the differences across genera (e.g, in terms of algal contact extent) also explained by differences in the sizes of corals?

line 267 – growth was not assessed

line 288 “process” is not appropriate here

line 292 – the Friedlander study refers to Caulerpa, not Bryopsis

figure 2 – I find this confusing and a but alarming – were sediments manually removed to create the present/absence treatmetns? Or were sediments visually assessed – the methods and this figure provide two different explanations.

figure 4 – What do the capital versus lower case letters for the SNK represent?

table 1 – what specific algae? I do understand what this plot is referring to
What does “Different algal contacts” mean? Is this referring to macroalgal or turf algal contact? Why is there a one way anova conducted in reference to the Turf+S and Turf-S? What are the specific algal contacts for the Kruskal Wallis tests?

Reviewer 2 ·

Basic reporting

Clear unambiguous professional English language used throughout: OK
The English grammar and spelling has improved considerably from the previous drafts. However, there are still some parts where the wording could be changed a bit to make things more easily understandable.

For instance the wording in the results section of the abstract is a bit hard to understand and could probably be stated better. More specifically, Lines 21-22 are a bit unclear. It might be better to say something like, Our data suggested that macroalgae was the main algal competitor for all corals, however, the extent of algal contact varied between coral genera. Although this sentence is still a little unclear, as it does not define what is meant by the "extent" of algal contact. You should also define what is meant by the proportion of interaction outcomes.

Introduction and background show context, literature is well-referenced and relevant: OK

The introduction is well-referenced and relevant. the only improvement I would suggest, is that when multiple factors or outcomes are discussed, a reference should be given for each factor unless the single reference given actually showed all of these different factors or outcomes. As an example I point to Lines 49-52. You should provide a reference for each factor (i.e., a reference for polyps size, colony size, growth form, and environmental factors).

structure conforms to PeerJ standards: Yes

Figures are relevant, high quality, well-labeled and describe: Yes

Figure 1: The caption references "the sheet". This is not a normal use of this terminology. It might be better to call it a table inset or something like that.

Figure 3: I would suggest not to use the * symbol to represent the data points outside of the 1.5 quartiles, as this symbol is typically used to denote significant differences. I would suggest using a square or triangle or really any symbol except for a *.

Figure 4: I believe "post hoc" should be italicized as it is a Latin phrase.

Figure 5: I once again, suggest not using the * symbol.

Raw data supplied: Yes

Original primary research within the scope of the journal: Yes

Experimental design

Methods described with sufficient detail and information to replicate: OK

Validity of the findings

Conclusions are well-stated, linked to the original research question, and limited to supporting results: OK

The discussion and conclusion sections have improved greatly, and are in general, well-stated, well-referenced, and supported by the research. The only small improvement I could suggest is that the last conclusion paragraph could be a little stronger and might end with more specific suggestions for follow-up questions and experiments rather than just saying that "experiments should be conducted".

Additional comments

Overall, the revised manuscript by Liao et al. continues to make improvements since it's last submission. The English grammar and spelling has improved considerably from the previous drafts. However, there are still some parts where the wording could be changed a bit to make things more easily understandable. The manuscript in its current form seems acceptable for publication pending some minor revisions.

---

## Round 0.4 · accepted · Accept

Thank you for working hard to improve the manuscript and making the most of the excellent work done by the reviewers to bring this manuscript up to the standards of PeerJ. Thank you for the opportunity to facilitate this publication. Best wishes - Craig

#